# Paper and screen media in current health education practices aimed at older adults: a scoping review protocol

Larissa Taveira Ferraz ,[1] Anna Julia Tavares Santos,[2] Lorena Jorge Lorenzi ,[3] Paula Costa Castro,[2] David Mark Frohlich,[1] Elizabeth Barley [4]

[1]Department of Music and Media, University of Surrey, Guildford, UK
[2]Department of Gerontology, Federal University of São Carlos, São Carlos, Brazil
[3]Interunits Graduate Program in Bioengineering- EESC/FMRP/IQSC, University of São Paulo, São Carlos, Brazil
[4]Department of Mental Health Sciences and Nursing, University of Surrey, Guildford, UK

**Correspondence to**
Larissa Taveira Ferraz;
l.taveiraferraz@surrey.ac.uk

## ABSTRACT

**Introduction** With technological advancement and the COVID-19 pandemic, paper-based media are giving way to screen-based media to promote healthy ageing. However, there is no review available covering paper and screen media use by older people, so the objective of this review is to map the current use of paper-based and/or screen-based media for health education aimed at older people.

**Methods and analysis** The literature will be searched in Scopus, Web of Science, Medline, Embase, Cinahl, The ACM Guide to Computing Literature and Psyinfo databases. Studies in English, Portuguese, Italian or Spanish published from 2012 to the date of the search will be examined. In addition, an additional strategy will be carried out, which will be a Google Scholar search, in which the first 300 studies according to Google's relevance algorithm will be verified. The terms used in the search strategy will be focused on older adults, health education, paper-based and screen-based media, preferences, intervention and other related terms. This review will include studies where the average age of the participants was 60 years or older and were users of health education strategies through paper-based or screen-based media. Two reviewers will carry out the selection of studies in five steps: identification of studies and removal of duplicates, pilot test, selection by reading titles and abstracts, full-text inclusion and search for additional sources. A third reviewer will resolve disagreements. To record information from the included studies, a data extraction form will be used. The quantitative data will be presented in a descriptive way and the qualitative data through Bardin's content analysis.

**Ethics and dissemination** Ethical approval is not applicable to the scoping review. The results will be disseminated through presentations at significant scientific events and published in journals in the area.

**Protocol registration number** Open science framework (DOI: DOI 10.17605/OSF.IO/GKEAH).

## INTRODUCTION

The world population continues to grow and age. This is driven by events such as the increase in life expectancy and the decrease in the fertility rate. According to United Nations data, the number of older adults will rise from 727 million people aged 65 and over in 2020 to more than double in three decades, about

### STRENGTHS AND LIMITATIONS OF THIS STUDY

⇒ A review that explores the use of both paper and screen as media for health education.
⇒ Seven databases will be consulted, as well as an additional strategy with the grey literature.
⇒ Literature in three languages besides English will be covered.
⇒ Only studies focusing on health promotion and prevention will be considered. Treatment studies will not be included, according to the purpose of this review.

1.5 billion in 2050. The projection is that they will transition from 9.3% to 16% of the population in these years.[1 2]

According to the 'Decade of Healthy Ageing: Plan of Action' document, the WHO defines healthy ageing as the process of 'developing and maintaining the functional ability that enables well-being in older age'. This functional ability is determined by the intrinsic capacity of the individual (physical and mental capabilities), also by the environment in which they live (physical, social and political) and the interaction among them.[3] Thus, efficiently managing health conditions through promotion and prevention practices has a direct influence on older adults' quality of life.[4 5]

Considering this perspective, health practices that involve promotion and prevention must be shaped by the contemporary context.[2] Therefore, many health education practices are aimed at disseminating information that stimulates changes in the health behaviour of individuals. Thus health communication is defined as 'the use of communication strategies (eg, interpersonal, digital and other media) to inform and influence decisions and actions to improve health'.[6] Such media can be paper-based, such as pamphlets and fliers, or screen-based, such as apps and websites.

Finally, it is important to note that with the current technological advance, paper-based media are giving space to screen-based media since with the COVID-19 pandemic and the rules of social distancing, there was a greater need to use these media.[7–9] Thus, m-health tools—apps for mobile devices—are being widely used for practices that help older adults to maintain or improve their health status.[10 11]

Although digital media are widely used in the current situation, there are some known barriers that make their use difficult. Digital media have to be accessed by digital devices that may be too expensive or difficult for older people to use, leading to digital exclusion. Furthermore, their information and interaction design may be difficult to read, understand or use due the use of small font sizes, long texts, no figures and disorganised layouts.[12] Paper media are more accessible than digital media but are potentially harder to disseminate to the right audience, are not easily updatable and lack multimedia illustration together with methods of gathering feedback. A combination of paper-and-digital media has been proposed recently through forms of 'augmented paper' with printed hotlinks that play on a nearby smartphone or TV.[13] This promises to couple the accessibility of paper with the updatability, interactivity and multisensory nature of digital media, effectively combining the best of both worlds for an older population.

To do this effectively, it is necessary to know what the best (and worst) aspects of both paper and screen media actually are for health promotion and education to the older population. This would also help health practitioners and publishers to improve the design information for each media separately and perhaps their selection for particular topics and audiences. Hence a scoping review of literature attending to the use of both paper-based and screen-based media for health would be a first step to mapping what is known about the advantages and disadvantages of both in a structured way, as this type of review has a broad objective and aims to map studies on a certain topic through a structured search.[14]

Previously, searches were conducted in Pubmed, Scopus and Google Scholar databases in order to verify the existence of reviews with this theme. From this search, it was identified that this would be a novel review with two related studies.[12 15]

The first study by Kampmeijer et al[15] is a systematic review of the literature that aims to distinguish the scope of use of e-health and m-health tools by participants aged over 50 years, as well as factors that influence this use. In this case, the study only addresses screen-based media, not paper-based, as well as focusing on primary prevention for people over 50 years of age. In addition, the review explicitly shows that in the last 4 years of the period covered by its analysis, they noticed a change in focus and a greater frequency of these tools, showing a rapid change among them, and new tools are expected to be available in the near future.

The second review by Zhao et al[12] is a scoping review that addresses the elderly and their main sources of health information on the internet but with a focus on factors that influence the behaviours of seeking this health information online by older adults. Thus, the article highlights the importance of screen-based media and technology in age-friendly design, both in usability and ease of use.

Both these reviews do not address the use of paper-based media in current forms of health education for older adults nor their comparison with screen-based media. In fact, a focus on *media* rather than technology use is a distinctive feature of our own approach not considered in previous reviews. Thus, the objective of the present review is to map the current use of paper-based and screen-based media for health education aimed at older adults, as well as to identify barriers and facilitators to the use of paper and screen health information.

## METHODS AND ANALYSIS

This protocol was developed following the guidelines of the Joanna Briggs Institute Revisions Manual (JBI)[16] and the instructions of *Preferred Reporting Items of Systematic Reviews and Meta-Analyses* (PRISMA) for Scoping Reviews, which was adapted due to being a scoping review protocol, taking into account only the items present in the protocol.[17] To develop the research, the PCC strategy was used: Population, Concept and Context. In this regard, the research questions are:

1. What and how are paper-based and screen-based media used in current forms of health promotion for older adults?
2. What are the barriers and facilitators to using paper-based and screen-based media for health information?
3. What are the differences between the use of paper and screen media for health education?

### Search strategy

At first, a preliminary search limited to Scopus, PubMed and Google Scholar databases was conducted to identify articles related to the topic and to gather keywords for this study. After this query, the following terms were defined: *older adults, health education, paper-based media, screen-based media, experience and intervention,* in addition to other related terms such as *print media, smartphone and tablet.*

Thereby, these terms serve as a basis for the development of a complete search strategy, which was adapted according to the characteristics of each database, using Boolean operators and MeSH terms as necessary. Such terms will be investigated in titles, abstracts and keywords of the studies. In brief, a preliminary search strategy for the Scopus database is presented in table 1.

The studies will be searched in Scopus (Elsevier), Web of Science (Clarivate), Medline, Embase (Elsevier), Cinahl (EBSCO), The ACM Guide to Computing Literature and Psyinfo (APA) databases. Studies in English, Portuguese, Italian or Spanish published from 2012 to the date of the search will be examined. Search strategies for all databases are described in online supplemental file

| Table 1 | Search strategy for scopus |
|---------|----------------------------|
| **Search** | **Keywords** |
| #1 | ('Older person' OR elder* OR 'older adults' OR 'elderly population' OR 'older people' OR ageing OR ageing OR 'older population' OR geriatric OR 'healthy ageing' OR 'successful ageing') |
| #2 | ('Paper-based media' OR 'screen-based media' OR website* OR platform* OR virtual OR online OR multimodal OR multimedia OR 'reading media' OR 'digital-based reading' OR 'paper based reading' OR flyer OR 'media advertisement' OR 'print media' OR app OR apps OR tablet* OR smartphone* OR m-health* OR e-health* OR 'patient information leaflets') |
| #3 | #3.1 ('health education' OR 'health information' OR 'health communication' OR 'health promotion') |
| | #3.2 (Preference* OR characteristic* OR experience* OR attribute* OR perception* OR development OR barrier* OR facilitator* OR opportunities OR problem* OR recommendations) |
| | #3.3 (trial OR intervention) |
| #4 | #1 AND #2 AND #3.1 AND #3.2 AND #3.3 |

1. In addition to these databases, an additional strategy will be carried out, which will be a Google Scholar search, in which the first 300 studies according to Google's relevance algorithm will be verified in order to include grey literature.

### Study inclusion criteria

For eligibility, scientific articles will be considered whose title and abstract have the objective or central research question with the presence of at least one health education intervention, whether on paper or screen, used to convey recommendations or guidelines aimed at promoting and protecting the health of older individuals.

This review will include studies that have a mean sample age of 60 years or older, of any gender or ethnicity, that are the target audience of health education strategies. These strategies must be focused on health promotion and prevention and not on treatments or care for pre-existing conditions through paper-based or screen-based media.

This scoping review will include articles with different study designs, whether quantitative, qualitative or mixed approach. For studies to be included in this review, they must be in English, Portuguese, Italian or Spanish, corresponding to the language skills of the reviewers. Also, it will include only articles that have been published in the last 10 years, from 2012 to the date of the search, given the popularisation of access to screen media, such as the use of smartphones, having then the coexistence of paper and screen media. Political documents and technical reports will not be included. Also, studies whose full texts cannot be found will be excluded after three attempts by email or phone to contact the authors.

It will also include, as an additional strategy, studies from the grey literature—those that are not officially published—covering theses, dissertations and opinion articles, given that the objective of a scoping review is to map studies, so the scope must be comprehensive.

### Study/source of evidence selection

The selection of studies will be done in five stages. We are currently carrying out the fourth stage. The first step was the bibliographic search in the chosen databases according to the specified search strategy. Then, the information was extracted into an Excel spreadsheet; after standardising all the articles in the same language, the duplicates were removed by two reviewers.

The second step was a pilot test, carried out to ensure a greater consistency among the reviewers. In this case, 25 studies were randomly selected from the search so that two reviewers independently verified their titles and abstracts, according to the eligibility criteria. There was a 92% agreement rate between the two reviewers, so the next step was started.

The third step was the reading of the titles and abstracts of all texts found in the search by the two reviewers, independently and double-blind (two reviewers read the text independently, without one reviewer being able to see the decision of the other), based on the eligibility criteria. The fourth step, which we are currently doing, is the reading of all the full texts of the studies accepted in the third step by the two reviewers independently to choose which ones will be included based on the inclusion criteria already established. If there is disagreement between the two reviewers in the third and fourth steps, the third reviewer will determine whether or not the study should be included.

The fifth and final step will be to carry out additional strategies, which will be the reading of the first 300 studies verified in Google Scholar according to their relevance algorithm, which may include the grey literature. This strategy will be executed according to the third and fourth step.

### Data extraction

A data extraction form with the objective of extracting and recording the main information from the included studies was developed. Two reviewers will verify the effectiveness of this form and if there is a need for any modification to it by extracting three studies. If there are any changes, they will be described in the full review manuscript.

There are recommendations from JBI that indicate specific details to be addressed in a data extraction form, such as study title, authors, year of publication, country of origin, population and quota sample, method and type of intervention.[16] Thus, the form developed for this review has these topics in addition to others related to the research questions, which address the characteristics of the sample, the central theme of health education, whether it is aimed to older adults or the caregiver, whether digital or paper media are used, characteristics of this media, as well as barriers and facilitators identified. To identify barriers and facilitators, all reports and narratives of successful experiences, positive perceptions, difficulties in use and impediments will be considered. The complete extraction form is described in online supplemental file 2.

### Data analysis and presentation
The data will be presented through graphs, figures or tables in order to identify, characterise and summarise evidence from the included studies based on the proposed objectives of this scoping review. In addition, a description of the study selection process and a PRISMA flowchart with the research results will be presented, as well as the excluded sources and the reason for this.[18] In addition, other data will be presented, such as the characteristics of the included studies and their samples, screen or paper media used and barriers and facilitators for the use of these media. Quantitative data will be presented descriptively and counted through relative and absolute frequency, mean and SD. Qualitative data will be analysed through Bardin's content analysis, in which the data will be allocated into categories that will be defined from the results coded according to similarities.[19]

The meaning of the results will be discussed in relation to the research questions of the scoping review described above (in Methods and analysis). For example, we will explain what kinds of paper and screen media have been used in health promotion to older adults and how (RQ1), the barriers and facilitators to improving them (RQ2), and the major differences in the way paper and screen media are used in this context. A likely date for the review to be completed and the final manuscript to be written and reviewed is June 2023.

### Patient and public involvement statement
None.

### ETHICS AND DISSEMINATION
Ethical approval is not applicable to this scoping review, as it only aims to gather information from published literature in the public domain. The results will be disseminated through presentations at significant scientific events and published in journals in the area.

**Acknowledgements** We would like to acknowledge University of Surrey Doctoral College, São Paulo Research Foundation (FAPESP) and Coordination for the Improvement of Higher Education Personnel (CAPES) for funding.

**Contributors** All authors have made substantial contributions to the development of the manuscript. LTF and AJTS conceptualised the research question and prepared the first draft of the manuscript. LTF, AJTS, LJL, PCC, DMF and EB contributed to the refining of the study design and guided the protocol development. Also, LTF, AJTS, LJL, PCC, DMF and EB contributed to the writing, editing and revising of this protocol. DMF assisted in the revision of the English.

**Funding** LTF received funding from the University of Surrey Doctoral College (Breaking Barriers Studentship Award), AJTS received funding from São Paulo Research Foundation (FAPESP), N° 2022/08918-6 to the development of this review and LJL received funding from Coordination for the Improvement of Higher Education Personnel (CAPES) - Financial Code 001.

**Competing interests** None declared.

**Patient and public involvement** Patients and/or the public were not involved in the design, conduct, reporting or dissemination plans of this research.

**Patient consent for publication** Not applicable.

**Provenance and peer review** Not commissioned; externally peer reviewed.

**ORCID iDs**
Larissa Taveira Ferraz http://orcid.org/0000-0003-3582-8097
Lorena Jorge Lorenzi http://orcid.org/0000-0002-2378-1263
Elizabeth Barley http://orcid.org/0000-0001-9955-0384

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
