## [Reviewer comments · BMJ Open]

ARTICLE DETAILS

TITLE (PROVISIONAL)	Paper and screen media in current health education practices aimed at older adults: a scoping review protocol
AUTHORS	Ferraz, Larissa; Santos, Anna Julia; Lorenzi, Lorena; Castro, Paula; Frohlich, David; Barley, Elizabeth

VERSION 1 – REVIEW

REVIEWER	Bas Geboers University Medical Center Groningen, Department of Health Sciences
REVIEW RETURNED	13-Feb-2023

GENERAL COMMENTS	This study protocol looks very good. The study addresses a relevant topic and the chosen design (a scoping review) seems to fit well with the aims of the study. The protocol is well written and is described in sufficient detail to be replicable. I only have a few minor queries, mostly for consideration for the final manuscript: - The introduction section could have been written in a somewhat more concise way. Especially the first page of this section (page 4) seems to contain more information than needed to support the relevance of the study. It is only on the second page that the key topics of the study are properly introduced.- I appreciate that the authors aim to include grey literature in the review. However, using the first 150 results from Google Scholar might not be the best possible way to achieve this. Most sources found via Google Scholar are published papers. A study has shown that most grey literature in Google Scholar is only found after page 20 of the results (see reference below). My suggestion would be to follow the recommendation by the authors of this paper and include the first 300 results from Google Scholar instead- I understand the decision to only focus on sources published in/after 2012. However, in 2012, the internet had already been around and played a major role in people's lives (including in their health) for some years. I would suggest to the authors to include any studies published after 2007 or 2002 instead, to better cover the full "digital era". It is likely that the number of sources identified from before 2012 is limited, so screening studies from between 2002/2007-2012 will probably not be that much extra work.-On page 8, line 51, the word 'double-blind' is used. It is not entirely clear what this means in this particular context. Could this be explained?-On page 10, February 2023 is mentioned as the intended month for writing the final manuscript. I wonder whether this is still accurate or whether this month should be changed to March/April/May, given the fact that it is already mid-February now.
---

	I am looking forward to reading the results of the scoping review. Haddaway NR, Collins AM, Coughlin D, Kirk S (2015) The Role of Google Scholar in Evidence Reviews and Its Applicability to Grey Literature Searching. PLoS ONE 10(9): e0138237. doi:10.1371/journal.pone.0138237
--	---

VERSION 1 – AUTHOR RESPONSE

Considering Reviewer 1, Dr. Bas Geboers, comments:

“This study protocol looks very good. The study addresses a relevant topic and the chosen design (a scoping review) seems to fit well with the aims of the study. The protocol is well written and is described in sufficient detail to be replicable.”

Answer: We appreciate the reviewer's comment.

“The introduction section could have been written in a somewhat more concise way. Especially the first page of this section (page 4) seems to contain more information than needed to support the relevance of the study. It is only on the second page that the key topics of the study are properly introduced.”

Answer: We appreciate the suggestion and we made changes to the introduction to summarize the relevant information. Thus, we have summarized information from the first page of the introduction through the following paragraph:

“According to the “Decade of Healthy Aging: Plan of Action” document, the World Health Organization defines healthy ageing as the process of “developing and maintaining the functional ability that enables well-being in older age”. This functional ability is determined by the intrinsic capacity of the individual (physical and mental capabilities), also by the environment in which they live (physical, social and political) and the interaction among them. Thus, efficiently managing health conditions through promotion and prevention practices has a direct influence on older adults' quality of life.”

“Considering this perspective, health practices that involve promotion and prevention must be shaped by the contemporary context. Therefore, many health education practices are aimed at disseminating information that stimulates changes in the health behaviour of individuals. Thus health communication is defined as “The use of communication strategies (e.g. interpersonal, digital and other media) to inform and influence decisions and actions to improve health.” Such media can be paper-based, such as pamphlets and fliers, or screen-based, such as apps and websites.”

“I appreciate that the authors aim to include grey literature in the review. However, using the first 150 results from Google Scholar might not be the best possible way to achieve this. Most sources found via Google Scholar are published papers. A study has shown that most grey literature in Google Scholar is only found after page 20 of the results (see reference below). My suggestion would be to follow the recommendation by the authors of this paper and include the first 300 results from Google Scholar instead”

Answer: We appreciate the suggestion and we will carry it out by reading the first 300 results from Google Scholar.

“I understand the decision to only focus on sources published in/after 2012. However, in 2012, the internet had already been around and played a major role in people's lives (including in their health) for some years. I would suggest to the authors to include any studies published after 2007 or 2002 instead, to better cover the full "digital era". It is likely that the number of sources identified from before 2012 is limited, so screening studies from between 2002/2007-2012 will probably not be that much extra work.”

Answer: We understand the reviewer's suggestion, however the aim of the review was to map the most recent studies, considering the first research question: "What and how are paper and screen-based media used in current forms of health promotion for older adults?". We want to see currently, because we have as one of our research questions to compare the use of paper and screen, "What are the differences between the use of paper and screen media for health education?"

In addition, we submitted the study to the BMJ Open on 28/09/2022, according to the newspaper's website they take an average of 59 days for the first decision, however, in our case, it took 170 days for the first decision, because of that, we had to start the review. We carried out the search in the databases, the pilot test and we have just finished reading the titles and abstracts. We are currently reading the full texts. In this way, we would not be able to modify the search strategy. We modified in the "Study/Source of evidence selection" session the current phase that we are in the review.

"On page 8, line 51, the word 'double-blind' is used. It is not entirely clear what this means in this particular context. Could this be explained?"

Answer: The "double-blind" expression means that two reviewers read the text independently, without one reviewer being able to see the decision of the other. Considering the possible doubt of the meaning, we chose to also clarify the term in the protocol.

"On page 10, February 2023 is mentioned as the intended month for writing the final manuscript. I wonder whether this is still accurate or whether this month should be changed to March/April/May, given the fact that it is already mid-February now."

Answer: Due to the delay in the first decision of the BMJ Open, mentioned previously, we started the review later than we had scheduled, so we changed the likely date for writing the final manuscript to June 2023 in the article.

VERSION 2 – REVIEW

REVIEWER	Bas Geboers University Medical Center Groningen, Department of Health Sciences
REVIEW RETURNED	18-Apr-2023
GENERAL COMMENTS	I appreciate the efforts made by the authors to process my comments. I agree with all changes made to the study protocol. Personally, I would have made a different choice with regard to the time span of the search (after 2002, instead of 2012). I am not fully convinced by the argument made by the authors that they would be unable to modify the search strategy in this stage; search engines include the option to search for sources published between 2002-2012. Nonetheless, from a pragmatic point of view, I understand the decision of the authors to not update their search strategy. Given that it is likely that the vast majority of potential relevant sources have been published after 2012, the decision to only include sources published after 2012 is unlikely to (strongly) alter the conclusions of the review and in no way disqualifies the review. I therefore support publication of this study protocol and I look forward to reading the results.

VERSION 2 – AUTHOR RESPONSE

We appreciate the careful review and comments regarding our study. We carried out the corrections and, in this letter, we clarify point by point, according to the requested changes:

Regarding the reviewer's, Dr. Bas Geboers, comment:

I appreciate the efforts made by the authors to process my comments. I agree with all changes made to the study protocol.

Personally, I would have made a different choice with regard to the time span of the search (after 2002, instead of 2012). I am not fully convinced by the argument made by the authors that they would be unable to modify the search strategy in this stage; search engines include the option to search for sources published between 2002-2012.

Nonetheless, from a pragmatic point of view, I understand the decision of the authors to not update their search strategy. Given that it is likely that the vast majority of potential relevant sources have been published after 2012, the decision to only include sources published after 2012 is unlikely to (strongly) alter the conclusions of the review and in no way disqualifies the review.

I therefore support publication of this study protocol and I look forward to reading the results.

Answer: We appreciate the comments made by the reviewer and agree that a search with a broader date would have slightly different results, however we also agree that most relevant studies on the researched topic are in the search date we chose. We appreciate the reviewer for agreeing with us and supporting the publication of the protocol.